# Space-time monitoring of groundwater fluctuations with passive seismic interferometry

Shujuan Mao [1,2,3] ✉, Albanne Lecointre [2], Robert D. van der Hilst [1] & Michel Campillo [1,2]

Historic levels of drought, globally, call for sustainable freshwater management. Under pressing demand is a refined understanding of the structures and dynamics of groundwater systems. Here we present an unconventional, cost-effective approach to aquifer monitoring using seismograph arrays. Employing advanced seismic interferometry techniques, we calculate the space-time evolution of relative changes in seismic velocity, as a measure of hydrological properties. During 2000–2020 in basins near Los Angeles, seismic velocity variations match groundwater tables measured in wells and surface deformations inferred from satellite sensing, but the seismological approach adds temporal and depth resolutions for deep structures and processes. Maps of long-term seismic velocity changes reveal distinct patterns (decline or recovery) of groundwater storage in basins that are adjacent but adjudicated to water districts conducting different pumping practices. This pilot application bridges the gap between seismology and hydrology, and shows the promise of leveraging seismometers worldwide to provide 4D characterizations of groundwater and other near-surface systems.

One of the worst droughts on record is ravaging the southwestern United States and regions elsewhere in the world. The severity and prolonged duration of this drought bring the water security to a tipping point. To address the water scarcity crisis requires efficient and sustainable management of freshwater resources. A critical component of freshwater is groundwater, which contributes over 60% of the total water supply in dry years in California. The state-of-health of regional groundwater systems has never been more important for agricultural, ecological, industrial, and urban purposes. However, owing to the remoteness and multi-scale, highly heterogeneous structures of water-bearing formations hidden beneath the Earth's surface, the spatial distribution of aquifers remains poorly understood. Equally limited is our knowledge of the temporal variability of groundwater storage on monthly to decadal timescales. The spatial and temporal constraints of aquifers are not only crucial for the efficient use of groundwater resources but also indispensable for avoiding permanent loss of storage capacity (which may occur when the pore pressure drops below pre-consolidation levels)[1–3]. Observations that characterize effectively the structures and dynamic behaviors of aquifers are thus imperative to sustain groundwater resources.

The traditional observation for tracking groundwater tables is the hydraulic head, which directly measures the water level of specific aquifer layers. In practice, however, the point-scale head measurements are too sparse in time or space to adequately capture variations of the highly nonuniform aquifer systems. Over the past decades, great progress has been made in hydrological investigations by utilizing a variety of non-invasive geophysical methods. Surveys exploiting the electromagnetic properties of the near-surface medium[4–8] allow to depict the groundwater systems with impressive detail, but they are expensive to conduct and often limited to static imaging on local scales. Remote sensing techniques have emerged as powerful monitoring tools to inform about the space-time variability of terrestrial

[1]Massachusetts Institute of Technology, Cambridge, USA. [2]Institut des Sciences de la Terre, Saint Martin d'Hères, France. [3]Present address: Stanford University, Stanford, USA. ✉e-mail: sjmao@stanford.edu

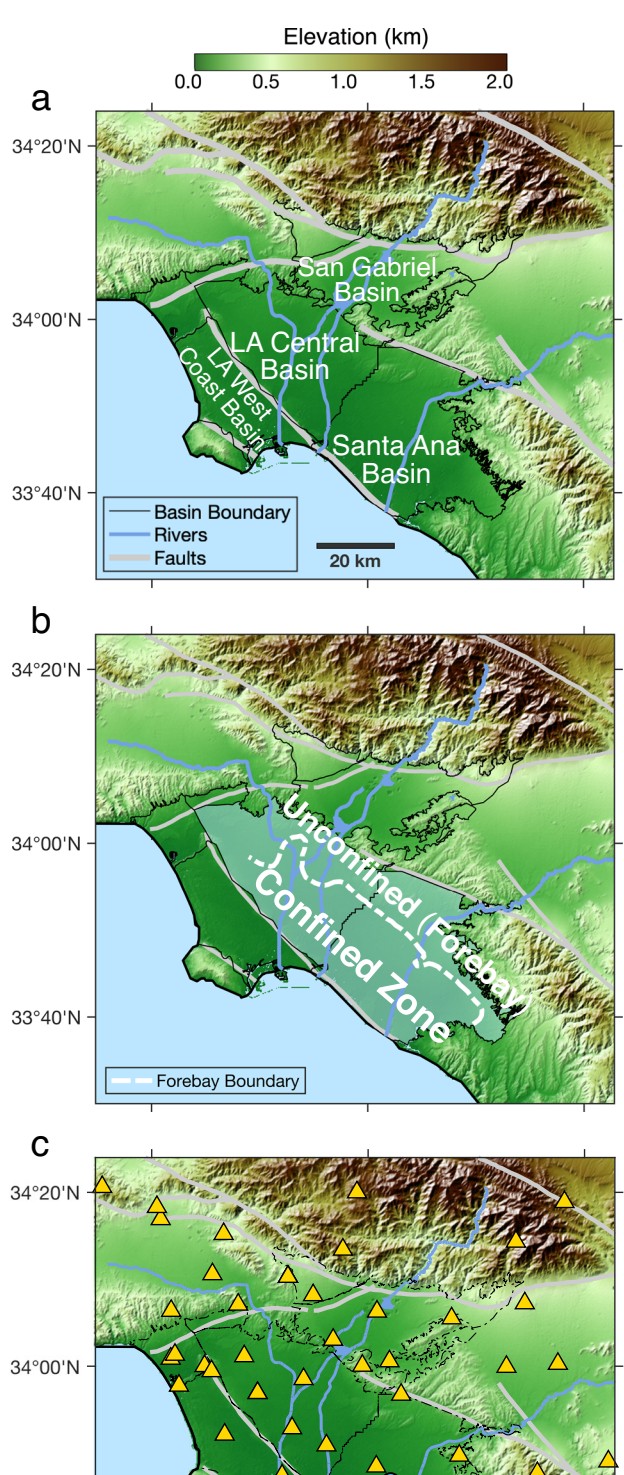

**Fig. 1 | Hydrogeological settings and station locations in the study area.**
**a** Groundwater basins near Los Angeles, California (adjudicated to different local water departments). Primary basins studied in this paper include the San Gabriel Basin, the Los Angeles (LA) Central Basin, the Santa Ana Basin, and the LA West Coast Basin. **b** The unconfined (forebay) and confined zones in LA Central and Santa Ana basins. The forebay boundary is estimated by the Orange County Water District (OCWD)[45] and the Water Replenishment District (WRD) of Southern California[46]. Confined to semi-confined conditions also exist in the San Gabriel Basin. **c** Station locations. Yellow triangles denote the broadband, 3-component seismic stations in the CI network operated by the Southern California Earthquake Data Center (SCEDC) available from 2000 to 2020. The blue circle denotes the groundwater monitoring well GGM-1/MP1 operated by OCWD, from which the hydraulic head data are shown in Fig. 2b. The violet diamond denotes the meteorological station ANA at which the precipitation record is shown in Fig. 2a.

Seismological observations can serve to add new constraints to the 4D (space-time) fluctuations of groundwater systems. Seismic velocity ($v$), or, rather, the propagation speed of seismic waves, is determined by elastic moduli and bulk density and can serve as an in-situ measure of the mechanical properties of the underground medium. In particular, relative changes in seismic velocity ($\Delta v/v$), which associate with changes in pore pressure, saturation, porosity, and micro-structures[20–23], can be used to infer the volume of stored groundwater. Modeling of seismic wavespeeds, taking into account the poroelastic effect and elastic loading[24], has suggested an anti-correlation between $\Delta v/v$ and groundwater level.

It has been demonstrated in recent years that $\Delta v/v$ can be measured continuously in time, with high sensitivity and at low cost, by combining two seismic interferometry techniques: (1) interferometry of seismic ambient noise fields, which provides repetitive estimates of Green's functions of the Earth interior structure[25–28], and (2) interferometry of seismic coda waves, which allows measurements of tiny changes in wavespeed[29,30]. This protocol has been employed to analyze time-evolving processes in various geological settings[31–38], including proof-of-concept studies for monitoring groundwater levels[31,39–42].

In this study, we present a 4D monitoring approach to push such seismological monitoring of groundwater (and other shallow sub-surface) systems to a new level, by not only inferring how $\Delta v/v$ changes in time (as in previous cases) but also in 3D space[43]. This advance is made possible by employing recently developed analytical descriptions of seismic coda propagation[44] in seismic interferometry. We apply our approach to the drought-stricken, densely populated Los Angeles (LA) coastal region in Southern California (Fig. 1) and measure the space-time evolution of $\Delta v/v$ during 2000–2020. Our results reveal the spatial patterns of seasonal and long-term fluctuations in groundwater volume at several hundred meters below the metropolitan LA[43].

## Results

### Seismic interferometry in Coastal Los Angeles Basins
We use records of seismic ambient noise from about 50 broadband seismic stations (Fig. 1c) in the Coastal Los Angeles Basins (CLAB) to calculate $\Delta v/v$. CLAB comprises a number of groundwater basins (Fig. 1a). The shallow subsurface is composed of unconsolidated Quaternary-age sediments mainly from marine or alluvial sources. These groundwater basins are mostly bounded by mountains and faults that can act as natural barriers of deep groundwater flow; they are also geographically partitioned by municipal borders (e.g., the boundary between Santa Ana and LA Central basins) and adjudicated to different local water departments.

The northwestern portion of LA Central and Santa Ana basins is characterized as an 'unconfined zone' or 'forebay'[45,46] (Fig. 1b), mainly consisting of aquifers made of coarser-grained sediments (sands and gravels) with high permeability. In contrast, the southwestern portion is characterized as a 'confined zone'[45,46], where lenses of aquifers are interbedded with laterally extensive, relatively impermeable lenses of

water storage. Yet they still have notable limitations. For example, satellite gravimetry[1,9] has a low spatiotemporal resolution (hundred-kilometer scale at monthly interval), and GPS and InSAR (Interferometric Aperture Radar) data resolve surface deformation but do not provide direct constraints on structures and processes beneath the surface[10–19].

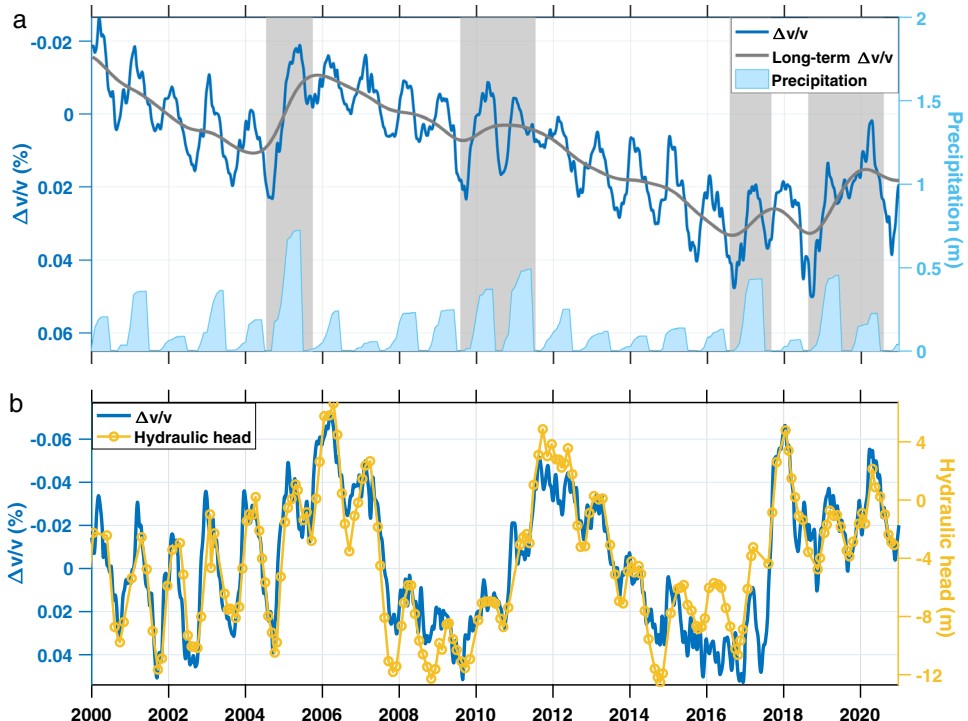

**Fig. 2 | Time series of $\Delta v/v$ (relative changes in seismic velocity), precipitation, and hydraulic head. a** The comparison between $\Delta v/v$ (bold blue line) and annual cumulative precipitation (light blue bars). $\Delta v/v$ is measured in 0.2–2.0 Hz and averaged over all spatial grids in the study area. The annual cumulative precipitation is measured at the meteorological station ANA (Fig. 1c). The gray line is obtained by low-pass filtering the blue line to show the long-term trend of $\Delta v/v$. The shaded areas (in gray) denote humid periods. **b** A local-scale comparison between $\Delta v/v$ (bold blue line) and the hydraulic head (yellow line with circles). $\Delta v/v$ is averaged over spatial grids within the Santa Ana Basin. Head data are measured at a dedicated groundwater monitoring well GGM-1/MP1 (Fig. 1c).

finer-grained sediments (silts and clays) (see Supplementary Note 1). Aquifers in confined to semi-confined conditions also exist in the San Gabriel Valley[17]. As rainfall is limited to winter and heavy well pumping mostly to dry summers, the groundwater storage in CLAB exhibits strong seasonal variability. In response to the seasonal groundwater fluctuations, confined aquifers show large deformation (leading to up to 60 mm annual uplift or subsidence at the ground surface[10,14–17]) due to the high compressibility of clays, whereas the deformation of unconfined aquifers is typically minute due to the low compressibility of sands and gravels.

Variations in seismic wavespeed are inferred from Green's functions obtained by cross-correlation (at different times) of seismic noise that is recorded continuously at stations in CLAB. We use a broad frequency band of 0.2–2.0 Hz and three sub-bands of 0.2–0.8, 0.4–1.6, and 0.5–2.0 Hz to compute time shifts along the seismic coda, which allows us to probe temporal changes at different depth[28,38,47,48]. To obtain the spatial distribution of $\Delta v/v$, we divide the study area into a 2 km-by-2 km grid (on the latitude-longitude plane) and invert for $\Delta v/v$ at each grid node for consecutive dates, based on the newly developed coda-wave sensitivity kernels (see, Methods subsection Imaging $\Delta v/v$ in space).

**Temporal fluctuations**
Figure 2a shows the $\Delta v/v$ time series from 2000 to 2020 (blue line) obtained from the 0.2–2.0 Hz band and averaged over all spatial grids in the study area. We plot $\Delta v/v$ in reverse direction for a more intuitive comparison with groundwater: Upwards corresponds to more groundwater and downwards to less groundwater. Also displayed is the long-term trend of $\Delta v/v$ (gray line) and the annual cumulative precipitation (light blue bars at the bottom).

Over the 21-year period, the blue line (that is, the opposite of $\Delta v/v$) in Fig. 2a exhibits a predominantly decreasing trend, suggesting an overall decline of groundwater associated with the long-term dearth of precipitation. The decreasing trend is interrupted by brief increasing

episodes, corresponding to the years with a few big storms. The blue line also exhibits seasonal fluctuations superimposed on the long-term trends of $\Delta v/v$: In each year, $\Delta v/v$ peaks around January–February and September with annual difference on the order of 0.01%. Featured by their phases and amplitudes, the seasonal variations of $\Delta v/v$ mainly stem from annual hydrological cycles[24]. These seasonal and long-term fluctuations will be analyzed separately in the following sections.

For examination on a more local scale, we focus on the Santa Ana Basin. Figure 2b shows $\Delta v/v$ averaged over spatial grids only within the Santa Ana Basin and the hydraulic head measured at a local monitoring well. This comparison demonstrates that during the time period under study, $\Delta v/v$ reflects the groundwater level regarding both seasonal fluctuations and long-term trends.

The good agreement between $\Delta v/v$ and the hydraulic head illustrates the promise of leveraging the large number of existing seismometers in California (and elsewhere worldwide) for tracking groundwater levels. This seismological alternative has three major advantages: First, $\Delta v/v$ can enhance considerably the spatial and temporal density of the in-situ well measurements, while alleviating the high cost of drilling and maintaining the dedicated monitoring wells. Secondly, $\Delta v/v$ is affected less by localized heterogeneity and more representative of the average hydrological conditions than the point-scale hydraulic head, because $\Delta v/v$ integrate the medium properties collectively along the pathways of coda waves. Thirdly, hydraulic head is not always proportional to groundwater storage, and most often the latter is the quantity of practical interests. In this regard, measurements of $\Delta v/v$ offer a new perspective (on mechanical properties of the aquifer medium), which can be integrated with the head data to constrain the groundwater volume.

**Imaging seasonal variations**
To investigate the spatial distribution of seasonal changes, we extract the amplitude of seasonal $\Delta v/v$ (Supplementary Fig. 1) in different

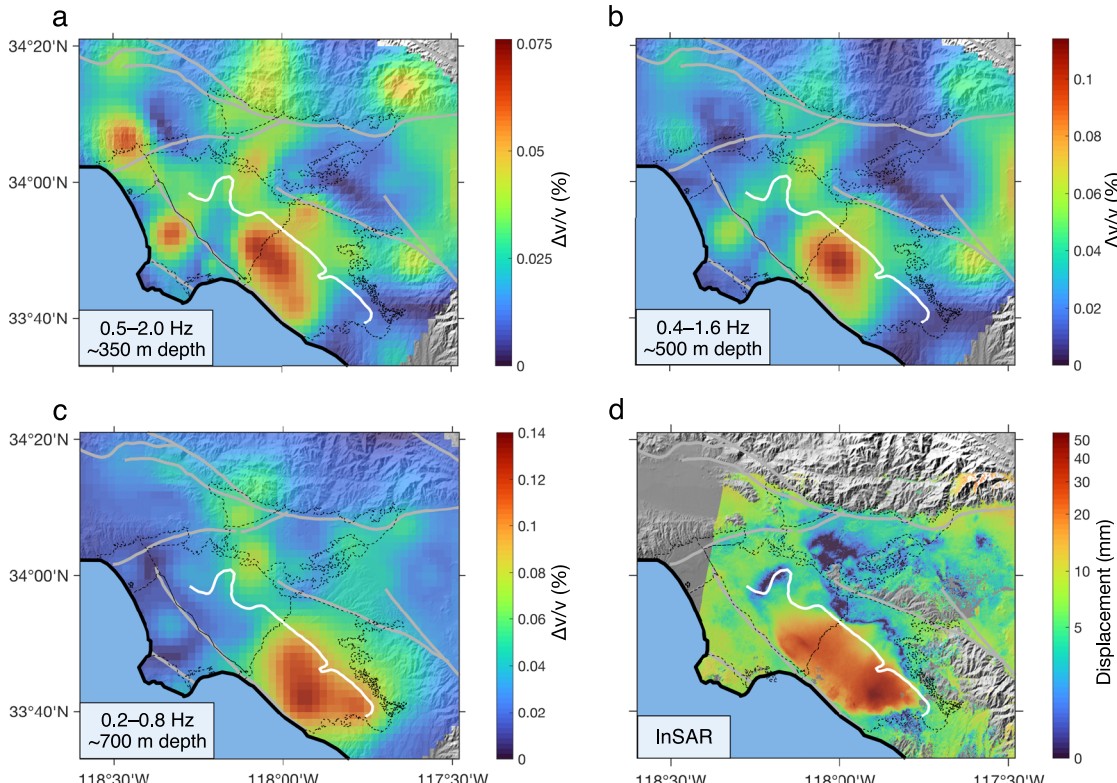

**Fig. 3 | Seasonal variabilities of $\Delta v/v$ and surface deformation. a–c** Maps showing the seasonal amplitude of $\Delta v/v$ measured in three decreasing frequency bands corresponding to increasing depths: **a** 0.5–2.0 Hz, corresponding to about 350 m depth; **b** 0.4–1.6 Hz, corresponding to about 500 m depth; **c** 0.2–0.8 Hz, corresponding to about 700 m depth. The depth sensitivities of seismic waves in different frequency bands are demonstrated in Supplementary Fig. 2. **d** The map of the seasonal amplitude of vertical displacement at the Earth's surface inferred from InSAR[10]. Warmer colors indicate stronger seasonal variability, and colder colors weaker seasonal variability. $\Delta v/v$ maps are derived using records from 2000–2011, and the InSAR map[10] during 1992–2011.

frequency bands (0.5–2.0 Hz, 0.4–1.6 Hz, and 0.2–0.8 Hz), which relate to medium properties at different depths (~350, 500, and 700 m, respectively) below the Earth's surface (see Methods subsection Depth sensitivity).

Figure 3a–c shows that the areas of large seasonal $\Delta v/v$ (in warm color) migrate southeastwards (from LA Central Basin to Santa Ana Basin) with increasing depth, suggesting that seasonal variations of aquifer storage deepen in that direction. Moreover, in each frequency (depth) range considered, the dominant seasonal $\Delta v/v$ occur in the confined zones of these basins, with sharp amplitude gradients along the forebay boundary. This indicates that the seasonal variations of hydrologic properties are much more significant in confined aquifers than in unconfined aquifers.

Seasonal variations can also be inferred from InSAR[10] (Fig. 3d), but the two types of data are different and complementary. InSAR measures the deformation at Earth's surface, which mainly manifests changes in pore pressure and thickness of aquifer layers. In contrast, $\Delta v/v$ measures the mechanical properties of crustal materials at depth, and thus reflects in-situ variations in pore pressure, saturation, mass density, and micro-structures. Measurements of surface deformation and $\Delta v/v$ have very different sensitivities to some particular types of subsurface changes. For example, saturation changes in vadose zone may be invisible to InSAR but can lead to considerable changes in $\Delta v/v$. Furthermore, $\Delta v/v$ is less affected by topography, weathering, or vegetation than InSAR.

Despite these differences, the maps of depth-dependent seasonal $\Delta v/v$ (Fig. 3a–c) and surface deformation (Fig. 3d) show largely consistent patterns of lateral variations, all with dominant changes in the confined zones of LA Central and Santa Ana Basins. The comparison between InSAR and $\Delta v/v$ maps confirms the expectation that InSAR-derived surface deformation reflects, to some degree, the integration of medium changes across different depths. Besides lateral variations on the surface, $\Delta v/v$ images further help to characterize the depth and thickness of the aquifer layers, which can be used by decision-makers in water agencies to optimize drilling and pumping strategies and, thus, to promote the efficiency and sustainability of groundwater management.

## Imaging cumulative variations

Across the study area, $\Delta v/v$ indicates an overall reduction of groundwater over the past two decades (Fig. 2a), reflecting a shift to a more arid climate. But in individual basins in CLAB, the long-term behaviors appear different (see Supplementary Fig. 3 for basin-wide $\Delta v/v$ time series). To assess the spatial variations of long-term changes, we map $\Delta v/v$ accumulated from 2000 to 2020 in Fig. 4a. It suggests that the San Gabriel and LA Central Basins stores less groundwater now than two decades ago; in the Santa Ana Basin, however, groundwater seems to recover with a slight increase than in the year 2000. In particular, the observations reveal opposite signs of cumulative groundwater change in adjacent basins (San Gabriel and LA Central versus Santa Ana), even though the only boundary between LA Central and Santa Ana basins is a county line, that is, a geopolitical boundary rather than a natural barrier for groundwater flow.

To explain the distinct cumulative patterns in adjacent basins, we analyze the relationship between the two major seasonal factors that influence the annual hydrologic cycles: the (natural) rainfall that recharges groundwater in winters and the (anthropogenic) well pumping that depletes groundwater in summers. The long-term groundwater change is primarily the year-to-year accumulation of annual differences between these two factors. In Santa Ana Basin (Fig. 4b), the linear relationship between pumping- and rainfall-induced

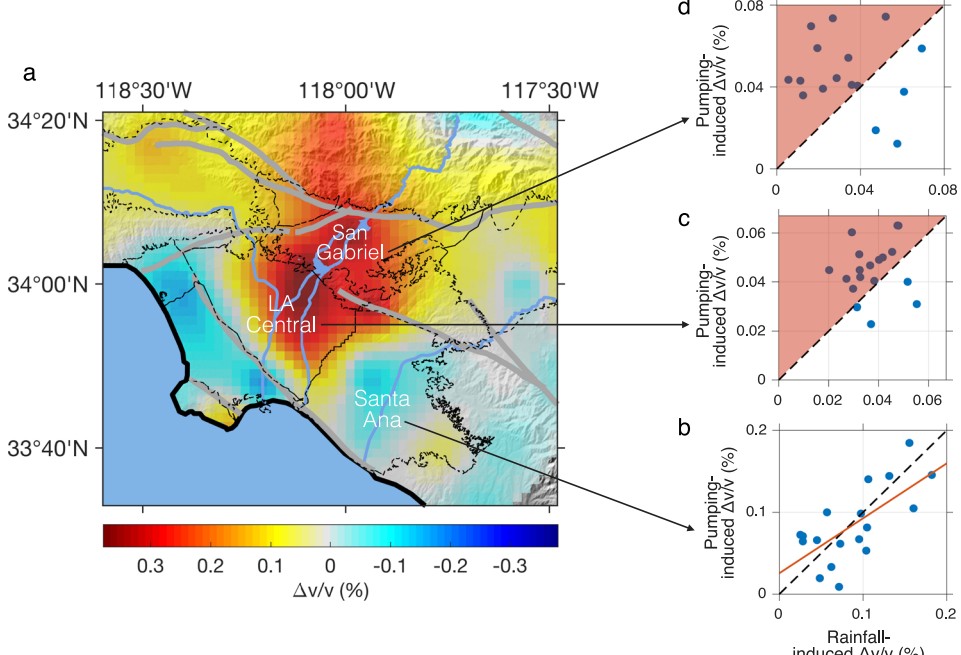

**Fig. 4 | Long-term $\Delta v/v$ in different groundwater basins. a** Map showing the $\Delta v/v$ accumulated from 2000 to 2020 (measured in 0.2–0.8 Hz). Colder colors associate with cumulative increases in groundwater and warmer colors with cumulative declines. Note that the boundary between the Santa Ana Basin and the LA Central Basin is a municipal border (i.e., the county line between LA and Orange counties) rather than a natural hydraulic barrier. See Supplementary Fig. 4 for the accumulated $\Delta v/v$ in 0.5–2.0 Hz (corresponding to shallower depth.) **b–d** Relationships between two seasonal components of $\Delta v/v$ (that is, the rainfall-induced decrease in winters and the pumping-induced increase in summers) in **b** Santa Ana Basin, **c** LA Central Basin, and **d** San Gabriel Basin. The rainfall-induced $\Delta v/v$ are averaged over three-year (backwards) moving windows to downweigh the effect of a few extreme climate events (following the protocol of OCWD[45]). The red line in b denotes the best-fit line (with a slope close to 1).

changes suggests that larger volumes of groundwater were pumped out in humid periods and less in arid periods and that in the long run the groundwater drawdown (by pumping) balances the recharge (by precipitation). As a result, total storage has been sustained. In contrast, in the LA Central and San Gabriel Basins (Fig. 4c, d), we find little evidence of a balance between groundwater withdrawal and replenishment. Instead, we observe more years when pumping-induced changes exceed rainfall-induced changes, suggesting that repeated groundwater overdraft leads progressively to excess depletion.

This comparison suggests that the pumping strategy of each municipal water district is the main reason for the distinct $\Delta v/v$ patterns in individual basins, and that anthropogenic activities, compounding the effect of climate change, play a significant role in shaping the hydrological systems. Effective management of groundwater resources is challenging in water-stressed regions worldwide due to a mixture of environmental, socio-cultural, economic, political, and historical factors. Nevertheless, the Santa Ana Basin case illustrates that putting into place sustainable pumping practices (which can now be verified and quantified via seismic interferometry) can indeed make a difference, setting an encouraging paragon to mitigate the effects of the warming climate.

The basin-wide seismological monitoring presented here can complement the traditional way of assessing groundwater storage (based on streamflow, precipitation, and well measurements), which is subject to uncertainties due to the lack of accurate aquifer models. This seismic approach provides an independent estimate of groundwater budget and safe yield and can thus aid in the policy-making in a sustainable manner.

## Discussion
This study presents an unconventional approach to the 4D monitoring of underground hydrologic processes, benefitting from our advances in seismic interferometry techniques. With a pilot application in CLAB,

we show that $\Delta v/v$ recovers the hydraulic head measured in groundwater wells during 2000–2020, illustrating its potential for enhancing the temporal and spatial density of isolated well measurements. Images of $\Delta v/v$ seasonality agree with surface deformation inferred from InSAR, but also enable the characterization of aquifer behaviors and hydrology at different depths. Based on assessment of long-term $\Delta v/v$, we find distinct patterns (decline or slight increase) of groundwater change accumulated in adjacent water districts, due to the effect of anthropogenic pumping practices compounding the climate change. This pilot application shows the promise of leveraging seismometers worldwide to monitor, image, and evaluate underground hydrologic processes. As an in-situ measure of subsurface material property, we anticipate $\Delta v/v$ to be a powerful 4D geodataset that will bring unique perspectives on constraining near-surface processes and assessing the ever-increasing impact of human activities on shaping the Earth's shallow subsurface. Integrated with other geodata, the seismological approach presented here can provide a comprehensive understanding of various environmental processes.

## Methods
### Passive seismic monitoring
The traditional way of passive seismic monitoring provides continuous measurements of $\Delta v/v$ (relative changes in seismic velocity). This approach incorporates the seismic interferometry in two steps: (1) interferometry of the continuous seismic ambient noise (2) interferometry of seismic coda waves. First, cross-correlating the continuous ambient noise recorded at two seismic receivers allows to estimate the Green's functions repetitively. Green's function is the impulse response of the medium at one receiver as if there were a source at the other, and it contains information about structures and elastic properties of the crustal medium between the two receivers[25,26]. Repeating noise interferometry at different calendar times yield estimates of Green's functions at consecutive dates. Second, interferometry of coda waveforms

gives measurements of $\Delta v/v$ between different dates. Coda waves are the late arrivals resulted from multiply scattered waves in the reconstructed Green's functions. By applying cross-spectrum analysis[29,47], coda-wave interferometry enables the measurements of tiny perturbations in seismic velocity ($\Delta v/v$ on the order of $10^{-4}$)[37,49,50]. These two techniques have been successfully applied together to study a wide range of time-varying crustal processes[31–38].

In this study, we apply the seismic interferometry using the existing, dense networks of broadband seismic stations in Los Angeles Metropolitan area (Fig. 1c) from the Southern California Earthquake Data Center (SCEDC). We make use of the continuous records of seismic ambient noise at all three components from about 50 stations during 2000–2020. The noise records are pre-processed by removal of instrumental response, whitening over 0.08–8.0 Hz in spectral domain and 1-bit normalization in time domain. We compute daily cross-correlations of pre-processed noise between all station pairs within 50 km and stack over 20 days. Then we apply interferometry between coda-waveforms from the cross-correlations at consecutive times (with 5-day step).

### Imaging $\Delta v/v$ in space

In the traditional framework of $\Delta v/v$ passive monitoring described above, the coda-wave interferometry is based on the assumption of homogeneous medium perturbation[29,30]. In other words, previous $\Delta v/v$ studies only consider the temporal variations, whereas in space the inhomogeneous distribution of $\Delta v/v$ are typically ignored and only averaged changes are estimated. However, in most monitoring applications, the spatial distributions of the perturbations are crucial for understanding the physical origin and mechanisms of the dynamic processes.

In this study, we go one step further to image the inhomogeneous $\Delta v/v$ distribution in space, by advancing the recent efforts to solve inverse problems[51–55] based on coda-wave sensitivity kernels[44,56–59]. The sensitivity kernels (Supplementary Fig. 5) describe the propagation of coda waves in a statistical sense by delineating the likelihood of the coda waves' travel path. Here we make use of the newly developed sensitivity kernels based on radiative transfer equation[44,59]. These new kernels provide more accurate descriptions of sensitivity taking into account the actual anisotropy of the scattered wave fields (compared to previous kernels derived under diffusion approximation[56,57]). Our measurements concern the early coda, which is primarily scattered surface waves[53,54,59] and are thus particularly important for $\Delta v/v$ imaging at shallow depth (e.g., the typical depth of groundwater aquifers).

The coda wave sensitivity kernels are then used to connect the spatial distribution of $\Delta v/v$ and travel-time shifts ($\Delta t$) between coda waveforms. In Eq. 1, the right-hand side is the spatial integration of $\Delta v/v$ at different spatial locations that are weighted by coda sensitivity kernel K, and the left-hand side is the $\Delta t$ measured at the corresponding lapse time. Note that here the sensitivity kernel is derived for two-dimensional infinite space[44] and the integration is conducted over the latitude-longitude plane.

$$\Delta t(t) = \int_S \frac{\Delta v}{v}(\vec{r}) \cdot K(\vec{r}, t; \vec{\mathscr{S}}, \vec{\mathscr{R}}) \cdot dS(\vec{r}) \tag{1}$$

Next, an inverse problem (Eq. 2) is set up and resolved to give the spatial distribution of $\Delta v/v$. The model parameters we invert for in vector **m** include $\Delta v/v$ at different spatial locations (Eq. 3); the observations, vector **d**, are the $\Delta t$ measurements at different lapse times between different station pairs (Eq. 4); and the design matrix **G** is constructed by coda-wave sensitivity kernels at the corresponding travel-times for all station pairs (Eq. 5). Supplementary Fig. 5 shows the sum of all kernels for all station pairs used in this study.

$$\mathbf{Gm} = \mathbf{d} \tag{2}$$

$$\mathbf{m} = \begin{pmatrix} \frac{\Delta v}{v}(\vec{\mathbf{r}}_1) \\ \frac{\Delta v}{v}(\vec{\mathbf{r}}_2) \\ \frac{\Delta v}{v}(\vec{\mathbf{r}}_3) \\ \vdots \\ \frac{\Delta v}{v}(\vec{\mathbf{r}}_n) \end{pmatrix} \tag{3}$$

$$\mathbf{d} = \begin{pmatrix} \Delta t_{11} \\ \Delta t_{12} \\ \vdots \\ \Delta t_{16} \\ \Delta t_{21} \\ \vdots \\ \Delta t_{26} \\ \Delta t_{31} \\ \vdots \\ \Delta t_{p6} \end{pmatrix} \tag{4}$$

$$\mathbf{G} = \begin{pmatrix} K_{11}(\vec{r}_1) & K_{11}(\vec{r}_2) & K_{11}(\vec{r}_3) & & K_{11}(\vec{r}_n) \\ K_{12}(\vec{r}_1) & K_{12}(\vec{r}_2) & K_{12}(\vec{r}_3) & \cdots & K_{12}(\vec{r}_n) \\ & \vdots & & \ddots & \vdots \\ K_{16}(\vec{r}_1) & K_{16}(\vec{r}_2) & K_{16}(\vec{r}_3) & \cdots & K_{16}(\vec{r}_n) \\ K_{21}(\vec{r}_1) & K_{21}(\vec{r}_2) & K_{21}(\vec{r}_3) & \cdots & K_{21}(\vec{r}_n) \\ & \vdots & & & \vdots \\ K_{26}(\vec{r}_1) & K_{26}(\vec{r}_2) & K_{26}(\vec{r}_3) & \cdots & K_{26}(\vec{r}_n) \\ K_{31}(\vec{r}_1) & K_{31}(\vec{r}_2) & K_{31}(\vec{r}_3) & \cdots & K_{31}(\vec{r}_n) \\ & \vdots & & \ddots & \vdots \\ K_{p6}(\vec{r}_1) & K_{p6}(\vec{r}_2) & K_{p6}(\vec{r}_3) & \cdots & K_{p6}(\vec{r}_n) \end{pmatrix} \tag{5}$$

Here we solve this inverse problem using a Bayesian approach[60] that maximizes the posterior probability assuming Gaussian distributions. In Eq. 6, $\mathbf{C_D}$ is the covariance matrix determined by the cross-correlation coefficient of the two small windows of coda waveforms for $\Delta t$ calculation, and $\mathbf{C_M}$ is the smoothing matrix that decays exponentially with distance.

$$\widetilde{\mathbf{m}} = \left(\mathbf{G}'\mathbf{C_D}^{-1}\mathbf{G} + \mathbf{C_M}^{-1}\right)^{-1}\mathbf{G}'\mathbf{C_D}^{-1}\mathbf{d} \tag{6}$$

The workflow of our data processing is summarized in Supplementary Fig. 6.

In this study, we divide the study area into 2 km-by-2 km grids (on horizontal plane) and calculate the coda-wave sensitivity kernels using a homogeneous model with scattering mean free path of 100 km. We then invert for the $\Delta v/v$ distribution on each grid. For each station pair, we compute $\Delta t$ at 6 lapse-time via coda-wave interferometry. Using $\Delta t$ calculated for all station pairs, the spatial inversions of $\Delta v/v$ are performed repetitively on consecutive dates (with 5-day steps) from 2000 to 2020. In the main text, we average the time series of $\Delta v/v$ over all grids in the study area (Fig. 2a) for an overall evaluation of temporal behavior; we also average over grids only within Santa Ana Basin (Fig. 2b) for a local-scale analysis.

## Depth sensitivity

To characterize the near-surface aquifers, in this study we use the early coda (with arrival time between 15 to 50 s), mainly consisting of scattered surface-wave content[52,53,58] that are most sensitive to shallow depth. As commonly used in surface wave studies, the seismic waves in different frequencies propagate and thus probe medium properties at different depth. Here we derive the depth sensitivity of $\Delta v/v$ measured in different frequency bands using kernels of Rayleigh waves[61] based on a 1D average velocity model[62] in Los Angeles basin (Supplementary Fig. 2a). As shown in Supplementary Fig. 2b, the three frequency bands, 0.5–2.0, 0.4–1.6, and 0.2–0.8 Hz, used for $\Delta v/v$ measurements are most sensitive to medium perturbations at around 350, 500, and 700 m beneath the surface.

## Data availability

The topographic data of metropolitan Los Angeles are available from the USGS Earth Resources Observation and Science (EROS) Center at https://www.usgs.gov/centers/eros/science/usgs-eros-archive-digital-elevation-shuttle-radar-topography-mission-srtm-1. The seismic data used in this study are available from the Southern California Earthquake Data Center (SCEDC) under network code CI for 2000 to 2020 at https://scedc.caltech.edu/data/waveform.html. The hydraulic head data are available from the Orange County Water District at https://www.ocwd.com/contact-us/public-records-request/utility-recordsrequest with well name GGM-1/MP1. The precipitation data can be downloaded from California Data Exchange Center, Department of Water Resources at https://cdec.water.ca.gov/snow_rain.html with station name ANA. The datasets of seismic velocity changes generated during the current study are available from the corresponding author on reasonable request.

## Code availability

The codes used in this study for the parallel computation of seismic time-shifts are publicly available online at https://gricad-gitlab.univ-grenoble-alpes.fr/lecoinal/doublets developed in Guix environment (https://guix.gnu.org).

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

## Acknowledgements

The authors thank Bryan Riel for offering the processed InSAR data from Riel et al.[10]. We thank Victor Tsai, Rosemary Knight, Thomas Herring, Mark Simons, and Jingyi Ann Chen for insightful discussions. We also thank Robert Clayton and Ellen Yu at SCEDC for providing the long-term seismic data. This paper is based on Chapter 3 of the first author's dissertation[43] at Massachusetts Institute of Technology. This work received funding from the European Research Council (ERC) under the European Union's Horizon 2020 Research and Innovation Program (N° 742335, F-IMAGE). S.M. acknowledges support from the George Thompson Fellowship at Stanford University.

## Author contributions

M.C. and S.M. designed this study. S.M. carried out the data processing and analysis, and wrote the initial manuscript. A.L. developed the parallel-computation codes for seismic time-shifts measurements. M.C. and R.v.d.H. supervised this project, and contributed to discussions of the results and improvements of the manuscript.

## Competing interests

The authors declare no competing interests.
