## [Peer Review File · Nature Communications]

REVIEWER COMMENTS

Reviewer #1 (Remarks to the Author):

Mao et al. present a novel analysis utilizing ambient noise records to characterize spatio-temporal variations in groundwater storage in Quaternary Basins in the Los Angeles region. While previous studies have shown that it is possible to track temporal variations with such data, to my knowledge this is the first study documenting the capability to image water level changes in space and as a function depth. The results are validated using independent well data and InSAR results, suggesting this methodology is quite accurate and in some cases superior to other observables. The approach and results suggest that it is in fact possible to track groundwater storage in space and time using advanced seismic interferometry techniques, which will likely be of value in a number of scientific and practical applications. The results are overall well documented and presented and the manuscript should be publishable with relatively minor revisions. I am returning a copy of the manuscript files with various edit suggestions and minor comments for the authors to consider. Below, are some comments that I hope the authors can fully address.

Line 123: Maybe I missed it, but I don't think it becomes clear earlier in the paper (~ line 65 etc.) how and in what sense dv/v is expected to vary with groundwater level.

Line 175: In addition to groundwater level effects, substantial deformation is also associated with oil fields in the area (e.g., see well locations and InSAR/GNSS deformation maps in Fig. 5 of Argus et al., 2004 doi:10.1029/2003JB002934). It seems worth considering any effects from this process on the results presented here. Apparently, based on section 3.2 in Argus et al., the relation between oil extraction and water extraction/injection in the fields (from <http://www.consrv.ca.gov/DOG/index.htm>) is rather complex.

Figure 1 and related text: There can still be quite substantial deformation (and confined conditions) to the N of the confined/unconfined dividing line (e.g., in San Gabriel Basin, Houlié et al., 2016 TGRS), so maybe reword this just a bit?

Figure 3 & 4: Stick to same characterization (either "warm colors" or "reddish colors" in both Fig. 3 and 4 captions. "warm/cold" seems more commonly used.

Author: roland1 Subject: Highlight Date: 4/5/22, 7:36:39 PM

Fig. 4: b-d may need more explanation in the caption to be clear. For example, in the supplement you mention a 3-year averaging scheme that should be mentioned and doesn't seem to quite make sense.

Supp21: Good to note depth range of the basins and their aquifers.

Supp65: Is this explanation clear? Consider rewording.

Supp75: Is this the amplitude of just the annual component?

Supp91: Are you referring variable behavior of basins in general, or opposite behavior of a particular pair of basins?

Supp98: The rain/pumping dominated periods are meant to overlap as indicated by the provided dates and be of unequal length?

Supp101: I don't really understand why this somewhat confusing, 3-year "backwards" averaging is needed. If it is, it needs to also be made very clear in the caption Fig. 4, not just here in the supplement.

Supp110: Can you please clarify what is meant by "artificial injection"? Does this refer to basin recharge using surficial recharge basins or actual pumping of water into the aquifer?

Supp122: Is there complementary evidence of seawater intrusion into the coastal aquifer? That should be a big concern.

Fig. S1: Can you put a scale on y-axis of the cross-section, even if it is somewhat approximate. What is known depth extent of basin fill from tomography, oil wells and/or geologic cross-sections?

Fig. S3: Can you please clarify if only the amplitude of the 1-year period is used to quantify seasonal amplitude?

Roland Bürgmann

Reviewer #2 (Remarks to the Author):

This paper presents an extension of dv/v methods applied to ambient noise in Southern California to track variations in aquifer levels related to pumping and recharge. An inverse methodology is proposed to create 4D maps of these variations, and the authors show that subtle phenomenology can be observed on multiple time scales related to aquifer pumping and recharge, and suggest that such permanent monitoring arrays could help dictate water use policy in the future.

This work is significant, less in terms of methodology though the authors do implement a novel inverse scheme, but more in terms of the societal impact aspect of it. Distributed seismic arrays could very much be used to adjust water use policy through such an approach.

The paper is robust in its explanations, data analysis, and interpretations.

The paper does however suffer from deficiencies in writing, where I have found a number of grammatical and sentence structure errors, and I would suggest a thorough revisiting of the text on that side of things.

Here are some examples in the first two pages:

32: the water scrutiny

32: "the" water scarcity crisis

53: "allow to depict"

54: "remote sensing has emerged as powerful monitoring tools"

56: Do not start a sentence with "but"

65: content, associated

66: rock damages

70: combing

92: "Their" refers grammatically to CLAB, not basins. "Subsurface" should be plural.

Structurally, the discussion section reads more like a conclusion, and all of the discussion is currently in the results section. I would suggest restructuring more cleanly.

Minor points:

-Although the inverse problem posed here is a novelty, I would downplay perhaps the statements on the novelty of radiative transfer sensitivity kernels. The cited papers are from ~2014, and it's been quite well known for some time (other works by Sato and Margerin, etc) that sensitivity kernels (and coda in general) are inherently related to diffusion and radiative transfer (with the latter accounting for ballistic contributions).

-Include map of stations used in this study in figure 1 rather than just the supplement.

-Calculating kernels requires a few inputs: what parameters did you use? Scattering mean free path? Absorption?

Overall, an excellent piece of work that, with a bit of cleaning up, will be an important contribution of the state of the art.

Point-by-point response to reviewers' comments

We appreciate very much the constructive and insightful reviews from the two reviewers. We carefully addressed the concerns of the reviewers and made corresponding clarifications or modifications. In our point-to-point response below, the reviewers' comments are *in grey* and our responses are *in blue* and are prefaced by "Response". In the revised manuscript, the changes we made are tracked *in red*.

Reviewer #1 (Remarks to the Author):

Mao et al. present a novel analysis utilizing ambient noise records to characterize spatio-temporal variations in groundwater storage in Quaternary Basins in the Los Angeles region. While previous studies have shown that it is possible to track temporal variations with such data, to my knowledge this is the first study documenting the capability to image water level changes in space and as a function depth. The results are validated using independent well data and InSAR results, suggesting this methodology is quite accurate and in some cases superior to other observables. The approach and results suggest that it is in fact possible to track groundwater storage in space and time using advanced seismic interferometry techniques, which will likely be of value in a number of scientific and practical applications. The results are overall well documented and presented and the manuscript should be publishable with relatively minor revisions. I am returning a copy of the manuscript files with various edit suggestions and minor comments for the authors to consider. Below, are some comments that I hope the authors can fully address.

Response: We thank the reviewer for the thorough edits, and have improved the manuscript based on these minor comments. Below are our responses to the main comments.

Line 123: Maybe I missed it, but I don't think it becomes clear earlier in the paper (~ line 65 etc.) how and in what sense dv/v is expected to vary with groundwater level.

Response: Thanks for pointing out this confusion. We modified line 67-69 to clarify this point in the introduction.

Line 175: In addition to groundwater level effects, substantial deformation is also associated with oil fields in the area (e.g., see well locations and InSAR/GNSS deformation maps in Fig. 5 of Argus et al., 2004 doi:10.1029/2003JB002934). It seems worth considering any effects from this process on the results presented here. Apparently, based on section 3.2 in Argus et al., the relation between oil extraction and water extraction/injection in the fields (from <http://www.consrv.ca.gov/DOG/index.htm>) is rather complex.

Response: Thank you for this suggestion. The ground deformations related to oil and gas operations in Los Angeles, as reported in previous studies^{9, 13-15}, are typically in much smaller scales (compared to the scale of groundwater aquifers in this area). The figure below shows an example of the bullseye-shaped uplift related to the

Santa Fe Springs Oil Field and subsidence related to the Wilmington Oil Field⁹, and both features are very localized.

From $\Delta v/v$ measurements in this study we didn't find such small-scale (less than ~ 5 km) features associated with oilfield. Our $\Delta v/v$ measurements may lack the areal spatial resolution to see them because of: (1) The large distance between seismic station pairs (~ 20 to 50 km); and (2) The smoothing and damping operations in the spatial inversion of $\Delta v/v$ (with 2 by 2 km grid size). So the spatial features at scales less than a few km can be averaged/smoothed out. We added a note about this point in the Supplementary Information (line 129-132).

Figure R1. Comparison of the surface deformation and $\Delta v/v$ regarding the oil/gas operations. *Left*: The surface displacement during 2004-2007 inferred from InSAR⁹. *Right*: $\Delta v/v$ accumulated during 2000-2020 (this is the Figure 4a in the main text). The spatial patterns on our $\Delta v/v$ map are in general consistent with the shapes of the groundwater basins; the small-scale features in the left figure related to the oilfields are not seen on the $\Delta v/v$ map.

Figure 1 and related text: There can still be quite substantial deformation (and confined conditions) to the N of the confined/unconfined dividing line (e.g., in San Gabriel Basin, Houlié et al., 2016 TGRS), so maybe reword this just a bit?

Response: Thanks for the comment. The deformation also occur above confined/semi-confined aquifers in San Gabriel. We modified the manuscript (Line 101-107 in main text, line 67-68 in Supp., and the caption for Fig. 1).

Figure 3 & 4: Stick to same characterization (either "warm colors" or "reddish colors" in both Fig. 3 and 4 captions. "warm/cold" seems more commonly used.

Author: roland1 Subject: Highlight Date: 4/5/22, 7:36:39 PM

Response: We modified the related texts and figure captions to consistently use 'warm/cold' description.

Fig. 4: b-d may need more explanation in the caption to be clear. For example, in the supplement you mention a 3-year averaging scheme that should be mentioned and doesn't seem to quite make sense.

Response: Thanks for the comment. The reason for our 3-year average of rainfall-induced changes is:

The Orange County Water Department (OCWD) manages the pumping in Santa Ana Basin by calculating the basin storage and setting a target amount of pumping on an annual basis⁵⁷. Then the OCWD allots the target pumpage to all the ‘Water Producers’ in the basin. For each producer, groundwater pumping above this allocation is assessed an additional charge that creates a disincentive for over-pumping.

The exact protocol by which OCWD determines the annual target pumpage is described as (quoted from the report of the water department⁵⁷): “*In any given year, groundwater withdrawals may exceed water recharged, as long as **over the course of a number of years** this is balanced by years when water recharged exceeds withdrawals.*”

The OCWD uses the groundwater recharge data over ‘a number of years’ instead of ‘in each year’, to give a more practical estimate by downweighing the effect of a few extreme climate events. We do not know exactly how many years are used by the OCWD to determine the target pumpage, so we choose a somewhat random but reasonable number, i.e., 3 years, in our evaluation. Nevertheless, our analysis based on $\Delta v/v$ shows that the OCWD does manage to keep the balance between the groundwater recharge and withdrawal over the course of every 3 years.

We modified the captions of Figure 4 and line 110-113 in Supp. to clarify this point.

Comments to Supplementary Information:

Supp21: Good to note depth range of the basins and their aquifers.

Response: We added this info in Fig. S1 and Supp. line 23-24.

Supp65: Is this explanation clear? Consider rewording.

Response: We improved this explanation in Supp. line 69-72.

Supp75: Is this the amplitude of just the annual component?

Response: This is the total amplitude of sinusoidal fitting (including 1- and 0.5-year periods).

Supp91: Are you referring variable behavior of basins in general, or opposite behavior of a particular pair of basins?

Response: We referred to increase of groundwater in Santa Ana versus decrease in LA Central and San Gabriel basins. We modified line 205-207 in the main text and line 99-100 in Supp. to clarify.

Supp98: The rain/pumping dominated periods are meant to overlap as indicated by the provided dates and be of unequal length?

Response: The rainfall/pumping-induced components are calculated as differences between peaks and valleys of dv/v in the time windows specified (rather than the difference of dv/v at the beginning and end of the time windows). We search for the peaks/valleys using relatively long windows (that overlap), because the peaks/valleys can occur at slightly different times in different years due to changes in the timing of rainfall and water demand.

Supp101: I don't really understand why this somewhat confusing, 3-year "backwards" averaging is needed. If it is, it needs to also be made very clear in the caption Fig. 4, not just here in the supplement.

Response: We explained for this point in the comment to Fig. 4 earlier.

Supp110: Can you please clarify what is meant by "artificial injection"? Does this refer to basin recharge using surficial recharge basins or actual pumping of water into the aquifer?

Response: We referred to both, including the groundwater replenishment by percolation or injection of water through the OCWD Forebay recharge basins and at injections wells of seawater intrusion barriers. The sources of these water include recycled water and purchased imported water from the Colorado River and California State Water Project. We modified the wording in Supp. line 121-123 to clarify this point.

Supp122: Is there complementary evidence of seawater intrusion into the coastal aquifer? That should be a big concern.

Response: Seawater intrusion has been a concern for LA West Coast Basin due to historical over-pumping, and multiple efforts are being made to monitor this issue. During the study period, according to the Report by the Water Replenishment District (WRD) of Los Angeles County⁶¹, the salt and nutrient concentrations monitored in 13 key wells in the West Coast Basin are generally stable, but a few individual well zones do show increasing trends.

Meanwhile, WRD has been implementing a set of programs to mitigate the saltwater intrusion, including injection of treated water at the seawater barrier wells. Benefiting from these artificial injections, groundwater levels at some monitoring wells show increasing trends during the study period, although some other wells still show decreasing trends (according to the Report by WRD⁶¹).

As discussed in the Supp. Section 4, based on $\Delta v/v$ measurements alone we cannot distinguish the changes of groundwater storage or density. Other types of measurements will be required to better understand this issue. We modified Supp. Section 4 a bit for this point.

Fig. S1: Can you put a scale on y-axis of the cross-section, even if it is somewhat approximate. What is known depth extent of basin fill from tomography, oil wells and/or geologic cross-sections?

Response: Based on the report of OCWD, the aquifer systems extend to a few hundred meters (up to about 700 m in the center of Santa Ana). We added the vertical scale on Fig. S1.

Fig. S3: Can you please clarify if only the amplitude of the 1-year period is used to quantify seasonal amplitude?

Response: Thanks for the comment. We used the sum of 1-year and 0.5-year sinusoids to determine the seasonal amplitudes. Including the 0.5-year period allows the best-fitting curve to have some distortions from the exact 1-year sinusoid, although the amplitude of the 0.5-year component is typically small (as shown by one example below.)

Figure R2. The seasonal fitting at one single grid in the study area. The amplitude of the component with 0.5-year period is typically small compared to the 1-year component (shown in bottom panel).

Roland Bürgmann

Reviewer #2 (Remarks to the Author):

This paper presents an extension of dv/v methods applied to ambient noise in Southern California to track variations in aquifer levels related to pumping and recharge. An inverse methodology is proposed to create 4D maps of these variations, and the authors show that subtle phenomenology can be observed on multiple time scales related to aquifer pumping and recharge, and suggest that such permanent monitoring arrays could help dictate water use policy in the future.

This work is significant, less in terms of methodology though the authors do implement a novel inverse scheme, but more in terms of the societal impact aspect of it. Distributed seismic arrays could very much be used to adjust water use policy through such an approach.

The paper is robust in its explanations, data analysis, and interpretations.

The paper does however suffer from deficiencies in writing, where I have found a number of grammatical and sentence structure errors, and I would suggest a thorough revisiting of the text on that side of things.

Here are some examples in the first two pages:

32: the water scrutiny

32: “the” water scarcity crisis

53: “allow to depict”

54: “remote sensing has emerged as powerful monitoring tools”

56: Do not start a sentence with “but”

65: content, associated

66: rock damages

70: combing

92: “Their” refers grammatically to CLAB, not basins. “Subsurface” should be plural.

Response: We thank the reviewer for pointing out the grammatical issues. In this revision we proofread the manuscript again and improved the writing.

Structurally, the discussion section reads more like a conclusion, and all of the discussion is currently in the results section. I would suggest restructuring more cleanly.

Response: Thanks for the suggestion. We restructure a bit the main text.

Minor points:

-Although the inverse problem posed here is a novelty, I would downplay perhaps the statements on the novelty of radiative transfer sensitivity kernels. The cited papers are from ~2014, and it’s been quite well known for some time (other works by Sato and Margerin, etc) that sensitivity kernels (and coda in general) are inherently related to diffusion and radiative transfer (with the latter accounting for ballistic contributions).

Response: Yes, it has been known for a while that coda waves can be described with the radiative transfer theory. However, (to our best knowledge) all previous studies based on travel-time sensitivity kernels used the ones under diffusion approximation, whereas the new travel-time kernels⁴¹ used in this study based only on radiative transfer equation. The novelty of the new kernel is that this formulation of the kernels itself takes into account the anisotropy of the scattered field by using specific (directional) intensity in place of energy density for the diffusion-based formulation. (Previous attempts have used the radiative-transfer equation to evaluate the energy density but kept the diffusion-based formulation of the kernel.) This is why we want to note the sensitivity kernel.

-Include map of stations used in this study in figure 1 rather than just the supplement.

Response: We moved Fig. S2 to Fig. 1c.

-Calculating kernels requires a few inputs: what parameters did you use? Scattering mean free path? Absorption?

Response: Thanks for the comment. The sensitivity kernels are calculated at 2 by 2 km grid with scattering mean free path of 100 km, following Margerin et al., 2016 (ref 40). In this study we use a homogeneous model without intrinsic absorption. These simplifications are shown to be reasonable to first order for the study area (proved by the $\Delta v/v$ results that are consistent with groundwater level and InSAR map). We modified the Method section line 345-347 to make this point clear.

Overall, an excellent piece of work that, with a bit of cleaning up, will be an important contribution of the state of the art.

REVIEWERS' COMMENTS

Reviewer #1 (Remarks to the Author):

I have read the revised manuscript, supplement and the response letter by the authors describing how reviewer concerns have been addressed. The authors have done a fine job doing so and I support publishing the paper in its current form. There are a few minor typos and grammatical issues, which I hope can be ironed out during the copy-editing stage. I am looking forward to seeing this fine contribution published soon.

Roland Burgmann

Point-by-point response to reviewers' comments

Reviewer #1 (Remarks to the Author):

I have read the revised manuscript, supplement and the response letter by the authors describing how reviewer concerns have been addressed. The authors have done a fine job doing so and I support publishing the paper in its current form. There are a few minor typos and grammatical issues, which I hope can be ironed out during the copy-editing stage. I am looking forward to seeing this fine contribution published soon.

Roland Burgmann

Response from the authors: We thank the reviewer for the constructive reviews. We have checked again the manuscript to correct for minor typos in this submission.